# Association between age of respiratory syncytial virus infection hospitalization and childhood asthma: A systematic review

Akihiro Shiroshita[1,2]*, Tebeb Gebretsadik[3], Pingsheng Wu[2], Nejla Zeynep Kubilay[4], Tina V. Hartert[2,5]

1 Division of Epidemiology, Department of Medicine, Vanderbilt University School of Medicine, Nashville, Tennessee, United States of America, 2 Division of Allergy, Pulmonary and Critical Care Medicine, Department of Medicine, Vanderbilt University Medical Center, Nashville, Tennessee, United States of America, 3 Department of Biostatistics, Vanderbilt University Medical Center, Nashville, Tennessee, United States of America, 4 Division of General Internal Medicine, Department of Medicine, Vanderbilt University Medical Center, Nashville, Tennessee, United States of America, 5 Department of Pediatrics, Vanderbilt University Medical Center, Nashville, Tennessee, United States of America

* akihiro.shiroshita@vanderbilt.edu

**Data Availability Statement:** All relevant data are within the manuscript and its Supporting Information files.

## Abstract

Identifying child age of RSV infection associated with increased risk of asthma is important for developing asthma prevention strategies. Our systematic review aimed to comprehensively summarize studies of the association between age of RSV infection and childhood asthma risk. The study protocol was pre-registered, and our study report adhered to the Preferred Reporting Items for Systematic Review and Meta-Analysis (PRISMA). Inclusion criteria were prospective and retrospective cohort studies and case-control studies which assessed the association of age of RSV infection before age 2 years and risk of childhood asthma after age two years. Relevant studies were identified through MEDLINE, Embase, Cochrane and International Clinical Trials Registry Platform (ICTRP) from study inception through May 5, 2023. Studies were evaluated with the Quality In Prognosis Studies (QUIPS) tool. From 149 studies screened, five studies (two prospective cohort studies and three retrospective cohort studies) were included in our systematic review, including 47,603 participants. Available studies only assessed age of severe RSV infection and asthma risk. The included studies used different age categories and outcome definitions, and were rated as having high risk of bias. Two studies had sample sizes of less than 300 and did not provide conclusive results related to age of RSV hospitalization and asthma risk. The other three studies reported RSV hospitalization between age 6 months and 23 months compared with age 0–6 months being associated with a higher odds ratio, hazard ratio, or incidence rate ratio of asthma diagnosis/hospitalization. Due to the heterogeneous epidemiological designs, including exposures and outcome ascertainments of the included studies, we could not perform a meta-analysis, or calculate weighted averages of the effect estimates. Our systematic review highlights a major gap in our knowledge about the relationship between age of RSV infection and asthma risk.

**Funding:** The author(s) received no specific funding for this work.

**Competing interests:** AS received financial support for his doctoral study from Vanderbilt University Medical Center, Center for Asthma Research and the Fulbright Association, TG is supported by grants from the NIH, TH is supported by grants from the NIH and serves on DSMBs for Pfizer, and as an external scientific consultant for Sanofi. this does not alter our adherence to PLOS ONE policy on sharing data and materials.

# Introduction

Asthma is the most common respiratory disease in children with an estimated prevalence of 10% [1]. Childhood asthma has been increasing over the past several decades, an increase that genetic factors alone cannot explain. Thus, identifying modifiable environmental risk factors and developing preventive interventions is critical to asthma prevention [2]. Among potentially modifiable risk factors, respiratory syncytial virus (RSV) infection early in life has been one of the most strongly and consistently associated risk factors for subsequent wheezing illness and asthma [3]. While delay of infection until after the first year of life (infancy) has a demonstrated protective effect, it remains unknown as to whether there is a critical age range during the first two years of life during which infection confers enhanced risk for subsequent childhood asthma inception [4].

There are multiple reasons to hypothesize that age of the RSV infection may differentially alter asthma risk. The risk for severe RSV infection in the first few months of life may be influenced by the infant level of maternal RSV antibodies, and the risk may increase over time as there is a gap between when maternal antibodies decline and infants develop a more mature immune system and airway [5]. Because lung alveolarization is not completed until age of 2–3 years, RSV infection before this age may have the greatest impact on airway development, and remodeling [6]. To our knowledge, no systematic review has been conducted to evaluate the association between age at RSV infection and childhood asthma inception. Determining whether an age-varying susceptibility to the long-term impact of RSV infection exists would advance our understanding of how early life viral infection contributes to respiratory morbidity, so that infection prevention or infection delay strategies could be implemented in susceptible populations [7]. Our systematic review comprehensively assessed peer-reviewed published articles that assessed age of RSV infection in the first two years of life with asthma risk. We additionally summarize the remaining gaps in our knowledge. The results should inform future research directions on this topic, identify key gaps in our knowledge, and potentially guide asthma prevention efforts.

# Materials and methods

## Study design

Our study is a systematic review and registered in advance (https://osf.io/awkgn/). We reported the study results according to the Preferred Reporting Items for Systematic Review and Meta-Analysis (PRISMA) [8]. Informed consent was not required due to the secondary data analysis design.

## Inclusion and exclusion criteria

Our research question in the PECO framework is P: child with and without an RSV infection <2 years of age, E: age at RSV infection before 2 years, C: other age at RSV infection before 2 years or no RSV infection before 2 years, O: childhood asthma determined after 2 years of age. Our search included observational studies (prospective and retrospective cohort studies, and case-control studies) evaluating the association between age of RSV infection and childhood asthma inception regardless of their methodologies. We excluded case series, case reports, review articles, and animal studies. In addition, we excluded published papers written in non-English languages and unpublished papers including pre-prints. To review as many relevant studies as possible, we used broad inclusion criteria. Study participants were not restricted based on medical conditions or preterm birth. The definition of RSV infection included a clinical diagnosis of early life RSV before 2 years, regardless of the laboratory confirmation. We

accepted an assumption of linearity on age. As for categorical age, if we could not extract information on the association of ≥1 age category, we excluded a study. We included studies that investigated associations between age of RSV infection and childhood asthma. The definition of childhood asthma included any clinical diagnosis of asthma after age two years [9]. We used the childhood asthma diagnosis defined in each study. The age ranges compared and the definitions of RSV hospitalization and asthma outcome for each study are detailed in Tables 2 and 3.

## Search strategy

Together with a health science librarian specialist, we used filters to reliably identify studies and search the following databases from their inception until May 3, 2023: MEDLINE (via OVID), Embase, Central, and International Clinical Trials Registry Platform (ICTRP). Our search terms are summarized in S1 Table. Recurrent wheeze was also used as a search term as it is a common precursor of asthma. In addition, we searched for other articles in the reference lists of the included articles and used the Web of Science citation search. AS screened articles for inclusion by reading the title and abstract. Then, through full-text review, AS determined the final articles for inclusion. Included articles were additionally reviewed by PW, NZK and TVH. For title and abstract screening and full-text review, we used Covidence systematic review software (Veritas Health Innovation, Melbourne, Australia).

## Data extraction

We extracted the following information from each study by reviewing full-text and supplementary materials: country, study design, hospital setting (e.g. community or academic hospital), patient inclusion and exclusion criteria, number of patients, patient demographic characteristics, definition of RSV hospitalization, determination and categories of age at RSV hospitalization, follow-up period, definition of childhood asthma inception (including age), statistical analysis, confounding factors adjusted in the analysis, effect size of age at RSV hospitalization on childhood asthma inception.

## Quality assessments

We performed quality assessment using the Quality In Prognosis Studies (QUIPS) tool [10]. It consists of six domains (i.e. study participation, attrition, prognostic factor measurement, outcome measurement, control of observed confounders, statistical analysis, and reporting). It assesses how each included study deviates from an ideal conceptual model. The conceptual model incorporated known confounding factors between age of RSV infection and childhood asthma inception [11, 12]. AS evaluated each study's quality based on the signaling question (S2 Table). PW, NZK, and TVH double-checked the validity of data extraction and quality assessment. If they had disagreements during the review process, they were resolved by discussion.

## Statistical analysis

Our initial plan was to perform a meta-analysis of the effect size of age of RSV infection on asthma risk. However, as there were no studies of RSV infection, we only included studies of age of RSV hospitalization (e.g. age <6 months vs. >6 months) on asthma risk within age categories with sufficient sample size and we provide weighted average estimates. However, due to heterogeneity in the study exposure (age of RSV hospitalization) and outcomes (asthma) as the included studies used different age categories, different asthma categories (e.g., asthma diagnosis and asthma hospitalization), and effect measures (e.g., regression coefficient and incident

rate), we were unable to perform a meta-analysis. Thus, we extracted the relevant data from the studies meeting the above-described criteria and provided the results with forest plots for the age categories of RSV hospitalization and asthma risk.

## Results

### Study selection

After screening 149 titles and abstracts, 21 studies proceeded to the full-text review; five studies met our inclusion criteria and are included in this systematic review (Fig 1) [13–17]. Tables 1, 2 and S3 Table summarize the study characteristics. All the included articles were prospective or retrospective cohort studies that evaluated age of RSV hospitalization. Koponen et al. and Muñoz-Quiles et al. included a range of bronchiolitis etiologies. Thus, we extracted the data only on RSV hospitalization. The three retrospective cohort studies, Muñoz-Quiles et al., Homaira et al., and Wang et al., utilized populations drawn from administrative databases.

### Quality assessments

We assessed the standard quality measures across studies based on the conceptual model (Fig 2), which are summarized below in S4 Table. The two prospective cohort studies, Koponen et al. and Zhou et al. had a high risk of bias in the domain of study attrition, outcome measurement, and confounding while the three retrospective cohort studies, Muñoz-Quiles et al., Homaira et al., and Wang et al., conducted using administrative data had a high risk of bias in study attrition, prognostic factor measurement, outcome measurement, and confounding (S4 Table).

Zhou et al. used a consecutive sampling and Koponen et al. did not report the sampling and recruitment scheme. These two prospective studies did not describe the detailed information on recruitment processes (e.g., number of screenings, approaches, and refusals). The other three retrospective studies used existing databases. Wang et al. used a matched-cohort design

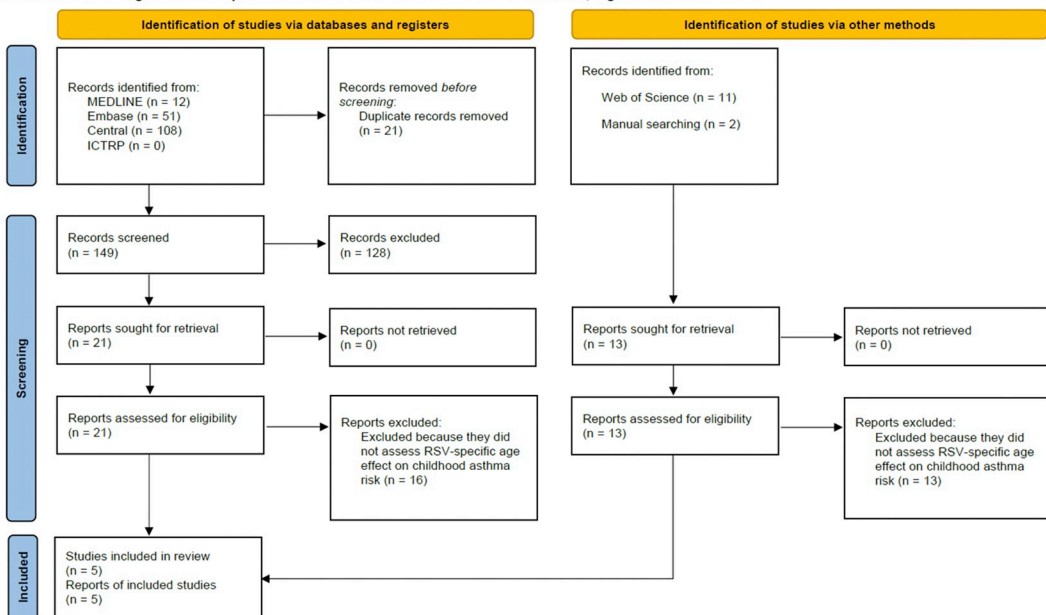

**Fig 1. PRISMA study selection flow diagram.** We screened a total of 187 studies (157 from MEDLINE, Embase, Central, or ICTRP, and 46 from citation search with Web of Science or manual searching). * ICTRP = International Clinical Trials Registry Platform.

Table 1. Study characteristics included in the systematic review of age of RSV hospitalization in the first two years of life and asthma risk after age two years.

| Study (first author and publication date) | Country | Study design and Number of participants | Setting | Inclusion criteria | Exclusion criteria |
|---|---|---|---|---|---|
| Muñoz-Quiles 2023 [13] | Spain | Retrospective cohort study (N* = 324,303) | Administrative database (Valencia Health System Integrated Database [VID]) | Children born between 2009 and 2015 residing in the Valencia region<br>Severe RSV previous bronchiolitis in children <2 years old: first RSV[†]-bronchiolitis hospitalization (ICD-9[‡] 466.11 or ICD-10[§] J21.0) or bronchiolitis hospitalization (ICD-9 466.1 or 466.19, or ICD-10 J21.8 or J21.9) in any diagnostic position with a laboratory-confirmed RSV result<br>No previous bronchiolitis <2 years old: children with no registers of bronchiolitis either at PC or hospital settings. | Born abroad or were not registered in the region due to lack of complete information<br><6 months' follow-up period<br>Registered with ≥42 days of age<br>Codification errors<br>Vaccinated with at least 1 dose of pneumococcal vaccine in private centers<br><1 year of follow-up after their second birthday |
| Koponen 2012 [15] | Finland | Prospective cohort study (N = 117) | University hospital | Hospitalized, full-term infants aged < 6 months due to bronchiolitis characterized by lower respiratory tract infection with rhinitis, cough and diffuse wheezes or crackles. | Not described |
| Homaira 2019 [16] | Australia | Retrospective cohort study (N = 18,042) | National database | RSV-coded first hospitalization (ICD-10 J12.1, J20.5, J21.0, and B97.4) before age 2 years | With ≥1 missing covariates |
| Zhou 2021 [14] | China | Prospective cohort study (N = 266) | University hospital | Hospitalized for RSV bronchiolitis as determined by clinicians, among children under 2 years old | Bronchopulmonary dysplasia<br>Cystic fibrosis<br>Interstitial lung diseases<br>Congenital heart diseases<br>Immunocompromised state<br>Co-infected with other pathogens |
| Wang 2022 [17] | United Kingdom | Retrospective cohort study (N = 23,365) | National database | RSV hospital admission (ICD-10 meeting both respiratory tract infection of J00-J06, J09-J18, J20-22 and RSV infection of J12.1; J20.5; J21.0; B97.4) in the first two year of life in the main or another 5 diagnosis fields<br>Control group included children born during the same period with unintentional accident hospitalization in any diagnosis field (V01-99, X00-59, X85-99, Y00-09, Y35-99) below 2 years of age | Asthma or wheeze admission before the start of follow up<br>Subjects with incomplete data on the covariates |

and children with RSV hospitalization were compared to those with unintentional accident hospitalization. Among the two prospective cohort studies, Zhou et al. excluded severe concomitant chronic diseases and Koponen et al. did not provide exclusion criteria [14, 15]. Three retrospective studies specifically evaluated the association of age of the first RSV hospitalization and risk of asthma [13, 16, 17]. All of the studies accounted for some confounding factors in their design and statistical analyses, but as with any observational study there could be remaining confounding.

Two retrospective cohort studies, Homaira et al. and Wang et al. used the first hospitalization due to asthma to define the asthma outcome, and Muñoz-Quiles et al. defined childhood asthma as ICD-10 code of asthma regardless of level of healthcare encounter [13, 16, 17]. Koponen et al. used a test of bronchial hyperresponsiveness as an objective criterion of asthma [15]. Zhou et al. used two separate outcome categories of asthma and recurrent wheeze. Regarding loss-to-follow-up, Zhou et al. described that there was no difference among children between those who were lost-to-follow-up and those who completed follow-up [14]. Koponen et al. included the reasons why children were lost-to-follow-up [15]. However, we were unable

**Table 2. Definition of exposure and outcome in five studies of the association of age of RSV hospitalization during the first 2 years of life and childhood asthma risk.**

| Study (first author and publication date) | Severe RSV* infection and age of hospitalization | Child asthma |
|---|---|---|
| Muñoz-Quiles 2023 [13] | RSV infection: First RSV-bronchiolitis hospitalization (ICD-9[†] 466.11 or ICD-10[‡] J21.0) or bronchiolitis hospitalization (ICD-9 466.1 or 466.19, or ICD-10 J21.8 or J21.9) with a laboratory-confirmed RSV result<br>Age: categorical, 0–5, 6–11, 12–17, and 18–23 months old | ≥3 recurrent wheeze-related ICD codes at least 1 month apart (ICD-9 466.XX, J21.X, J20, or ICD-10 R0.62, J98.01, 519.11, or 786.07) within a year or at least 1 asthma code (ICD-9 493.XX, or ICD-10 J45, J46, J45.9, J45.8, J45.1, or J45.0) at hospital or primary care between 2 and 4 years |
| Koponen 2012 [15] | **RSV infection:** Hospitalization with wheezing due to RSV detected in nasopharyngeal aspirates by immunofluorescence<br>**Control:** None<br>**Age:** categorical, <3 months and 3 months to <6 months | Current asthma: Children between 5 to 7 years<br>• Using continuous maintenance inhaled corticosteroids for asthma, or<br>• Suffering from doctor-diagnosed wheezing or prolonged (>4 weeks) cough or night cough, apart from infection, during the preceding 12 months in addition to bronchial hyperresponsiveness in exercise challenge test |
| Homaira 2019 [16] | **RSV infection:** First RSV-coded hospitalization (ICD-10 J12.1, J20.5, J21.0, and B97.4) before age 2 years<br>**Age:** categorical, <3 months, 3 months to <6 months, 6 months to <12 months and 12 months to ≤24 months | First asthma hospitalization with asthma-specific codes (ICD-10 J45, J46) after age 2 years |
| Zhou 2021 [14] | **RSV infection:** RSV hospitalization confirmed by nasal or oropharyngeal swabs with RSV detected by direct immunofluorescence<br>**Age:** continuous age under 2 years old | Asthma defined based on the Global Initiative for Asthma (GINA) recommendation (no further details) at 1, 3, 5 and 7 years after RSV hospitalization discharge |
| Wang 2022 [17] | **RSV infection:** First RSV hospitalization (ICD-10 meeting both respiratory tract infection of J00-J06, J09-J18, J20-22 and RSV infection of J12.1; J20.5; J21.0; B97.4) in the first two year of life<br>**Control:** Children born in the same period of time and had an unintentional accident hospitalization (V01-99, X00-59, X85-99, Y00-09, Y35-99) below age 2 years<br>**Age:** categorical, < 6 month and 6–23 months | Asthma or wheeze admission at least 4 weeks after the index hospitalization event, with follow-up periods evaluated at 0-< 2 years, 2-<4 years, 4-<6 years, and ≥ 6 years (ICD-10 J45, J46, R06.2) between January 2001 and 2016 |

Abbreviations: *:RSV = respiratory syncytial virus, †: ICD-9 = 9th revision of the International Statistical Classification of Diseases and Related Health Problems, ‡: ICD-10 = 10th revision of the International Statistical Classification of Diseases and Related Health Problems

to determine if there were baseline differences between children who were lost-to-follow-up and those who were included in the study by Koponen et al [14, 15].

## Association of age at RSV hospitalization and risk of childhood asthma

Fig 3 and Table 3 summarize the findings from the five studies and provide the effect sizes of age of RSV hospitalization on asthma risk across the studies. Muñoz-Quiles et al. compared age-stratified odds of recurrent wheeze/asthma at age 2–4 years between those with RSV hospitalization and those with no bronchiolitis during the first 2 years of life. Thus, we calculated unadjusted odds ratio comparing two age categories (Table 3). Older age categories had a higher odds ratio of asthma compared to younger age categories. In the study by Koponen et al. they presented raw data on age of hospitalization for RSV bronchiolitis and asthma inception, but the multivariable logistic regression model controls for the status of "non-RSV bronchiolitis" (yes vs. no) [15]. Thus, we could not extract an RSV-specific age effect for this study. In the study by Zhou et al. the age effect of RSV hospitalization on asthma risk was assumed to be linear from 0 to 24 months and three outcomes (no cases, recurrent wheeze, and asthma) were simultaneously evaluated by using a multinomial logistic regression. Recurrent wheeze and asthma were exclusively evaluated in the same multinomial logistic regression. Age of RSV hospitalization was significantly associated with recurrent wheeze (adjusted odds ratio = 1.42; 95% confidence interval [CI], 1.01 to 2.01, p-value = 0.05), but not with asthma

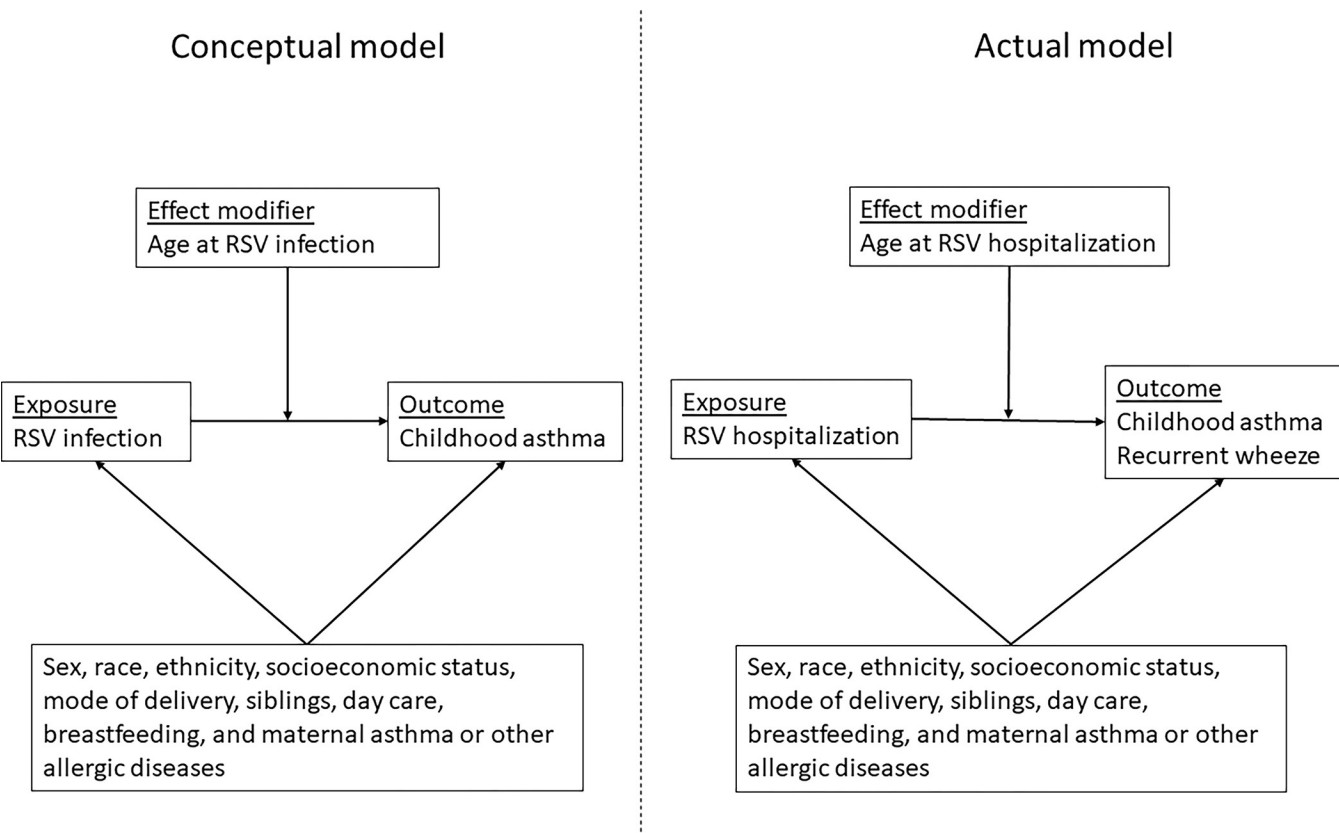

**Fig 2. Ideal conceptual model of the relationship between age of RSV infection and childhood asthma risk (left panel), and the actual model of RSV hospitalization and childhood asthma risk (right panel) of our systematic review.** * RSV = respiratory syncytial virus. Our original conceptual model was 'age of RSV infection during the first two years of life', however, there were no studies of the effect of age of RSV infection during the first two years of life on asthma risk, so we included studies which assessed age of RSV hospitalization instead.

(adjusted odds ratio = 1.27; 95% CI, 0.96 to 1.67; p-value = 0.10). Homaira et al. compared the incident rate of first asthma hospitalization among children with an RSV hospitalization at <3 months and older age categories (i.e., 3 to <6 months, 6 to <12 months, and 12 to ≤24 months). In this study RSV hospitalization at age <3 months had the lowest risk of having an asthma hospitalization after age 2 years (Fig 3). The study by Wang et al. estimated the hazard ratios of first asthma hospitalization among those with RSV hospitalizations at age 0–5 months and 6–23, compared to those without RSV hospitalization. Children with RSV hospitalization at age 6–23 months had a significantly higher hazard ratio of asthma hospitalization during age 4–6 years (adjusted hazard ratios: 1.9 [95% CI: 1.2–3.0, p-value = 0.01]) while RSV hospitalization at age 0–5 months was not (adjusted hazard ratios: 1.0 [95% CI: 0.6–1.8, p-value = 0.92]). The rate of first asthma hospitalization was higher among children with RSV hospitalization at age 6–23 months compared to those at age 0–5 months (3.5 rates/1000 child-years [95% CI: 2.6–4.6] vs. 2.8 rates/1000 child-years [95% CI: 2.1–3.6]).

## Discussion

### Principal findings

Given the challenges of conducting RSV surveillance and the paucity of studies meeting our criteria we were unable to accomplish the goals of our conceptual model, thus our systematic

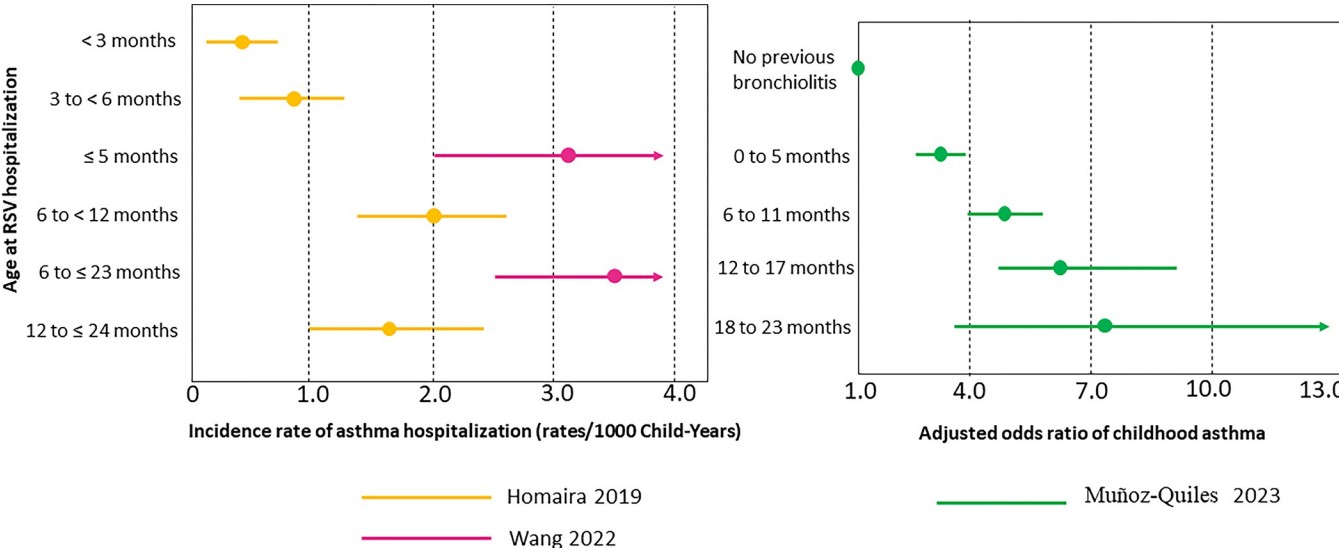

**Fig 3. Forest plots of the association of age of child RSV hospitalization on asthma risk in each study (color-coded) included in the systematic review for which we could extract summary statistics.** Homaira et al. directly compared the incident rate ratio of asthma hospitalization between RSV hospitalization at <3 months and older age categories (i.e., 3 to <6 months, 6 to <12 months, and 12 to ≤24 months). Age <3 months had the lowest risk of asthma hospitalization (incidence rate ratio: 3 to <6 months vs. <3 months, 1.9 [95% CI: 0.1–3.7]; 6 to <12 months vs. <3 months, 4.4 [95% CI: 2.5–7.8]; 12 to ≤24 months vs. <3 months, 3.8 [95% CI: 2.0–7.2]. Wang et al. described first asthma and wheeze admission rates of age ≤ 5 months and 6–23 months, compared to unintentional accident hospitalization in 6 years or older (per 1000 child-years). Muñoz-Quiles et al. compared odds ratio of childhood asthma between RSV hospitalization in age categories and no previous bronchiolitis.

review focused on the available studies of the association of age of RSV hospitalization in the first 2 years of life and subsequent risk of childhood asthma (Fig 1). These studies suggest that there is a trend in RSV hospitalization after 6 months of age being associated with an increased risk of asthma diagnosis/hospitalization compared to RSV hospitalization between 0–6 months. However, based on the data available we were unable to identify a critical age at which RSV infection confers enhanced risk of later childhood asthma nor determine the association between age of RSV infection and childhood asthma.

## Strengths and limitations

To our knowledge, this is the first systematic review that aims to define an early life age range or critical susceptibility period during which RSV infection is associated with the highest risk of childhood asthma inception. Our search identified no studies that investigated the association between age of RSV infection during the first two years of life and asthma risk; instead all identified studies focused on age of RSV hospitalization and asthma risk. Due to heterogeneity in age categories and asthma definitions among studies, we were unable to perform a meta-analysis. Studies of severe RSV infection may be biased as a less severely ill young infant is more likely to be hospitalized compared with an older infant, such that hospitalization of older infants and children may identify children with more severe RSV disease, demonstrating a severity-dependent rather than age-dependent association. Additionally, mild cases of RSV infections regardless of age may not be tested in the outpatient setting, which further adds to potential of missing the exact number of cases at a certain age interval. An additional limitation is that the interval between RSV hospitalization and asthma diagnosis/hospitalization was not clearly defined in all studies, recognizing that not all of these studies were designed to assess the association of age of infection and asthma risk. While Koponen et al. defined RSV hospitalization up to 6 months of age and asthma between 5 and 7 years of age, other studies

**Table 3. Summary of statistical analyses and estimated effect of age of RSV hospitalization on child asthma inception.**

| Study (first author and publication date) | Statistical analysis | Confounding factors | Odds ratio/hazard ratio of RSV* age-effect on child asthma inception | Proportion/incidence rate of child asthma |
|---|---|---|---|---|
| Muñoz-Quiles 2023 [13] | Logistic regression with age stratification | Prematurity<br>Sex<br>Antibiotic consumption in the first year of life Vaccination against pneumococcus<br>Health department<br>Year and month of birth | The adjusted odds ratio of recurrent wheeze/asthma between 2 and 4 years old (208,490 children with no bronchiolitis as the reference)<br>• Age 0 to 5 months (213,698 participants): 3.07 (95% CI†: 2.88−3.27)<br>• Age 6 to 11 months (208,926 participants): 4.69 (95% CI: 3.88−5.68)• Age 12 to 17 months (208,589 participants): 5.41 (95% CI: 3.59−8.14)• Age 18 to 23 months (208,530 participants): 7.07 (95% CI: 3.75−13.34)<br>Unadjusted odds ratio of wheeze/asthma between 2 and 4 years comparing two age ranges• Age 0 to 5 months vs. 6 to 11 months: 0.66 (95% CI: 0.54−0.81)• Age 0 to 5 months vs. 12 to 17 months: 0.62 (95% CI: 0.41−0.94)• Age 0 to 5 months vs. 18 to 23 months: 0.33 (95% CI: 0.18−0.62)• Age 6 to 11 months vs. 12 to 17 months: 0.40 (95% CI: 0.26−0.62)• Age 6 to 11 months vs. 18 to 23 months: 0.50 (95% CI: 0.26−0.96)• Age 12 to 17 months vs. 18 to 23 months: 0.53 (95% CI: 0.25−1.12) | Not applicable |
| Homaira 2019 [16] | Poisson regression | Parity of the mother<br>Smoking<br>Indigenous status<br>Sex<br>Hospitalization due to other respiratory viruses in the first 2 years of life | Not applicable | The adjusted incidence rate of first asthma hospitalization<br>• Age <3 months (6,158 participants): 0.5 events/1000 Child-Years (95% CI: 0.2−0.7)<br>• Age 3 months to <6 months (4,110 participants): 0.9 events/1000 Child-Years (95% CI: 0.5−1.3)<br>• Age 6 months to <12 months (4,540 participants): 2.0 events/1000 Child-Years (1.4−2.7)<br>• Age 12 months to ≤24 months (3,234 participants): 1.7 events/1000 Child-Years (1.0−2.5) |
| Koponen 2012 [15] | Multivariable logistic regression incorporating age as a covariate | Gender<br>Atopic dermatitis at <12 months of age<br>Non-RSV bronchiolitis<br>Maternal history of asthma | Odds ratio of age of 3 months to <6 compared to <3 months: 2.04 (0.64−6.04) | Not applicable |
| Zhou 2021 [14] | Multivariable and multinomial logistic regression incorporating age as a covariate | Gender<br>Cesarean section<br>Preterm birth<br>Birth weight<br>First child<br>Eczema<br>Family allergy<br>Severity<br>Fever<br>Wheezing duration | Age (0–24 months, continuous, 266 participants): adjusted odds ratio for asthma, 1.27 (95% CI: 0.96–1.67); that for recurrent wheeze, 1.42 (95% CI: 1.01–2.01) | Overall proportion of asthma, 65/266 (24.4%); that of recurrent wheeze, 165/366 (45.1%) |

*(Continued)*

**Table 3.** (Continued)

| Study (first author and publication date) | Statistical analysis | Confounding factors | Odds ratio/hazard ratio of RSV* age-effect on child asthma inception | Proportion/incidence rate of child asthma |
|---|---|---|---|---|
| Wang 2022 [17] | Cox regression with interaction with RSV infection and time | Sex<br>Quantiles of Scottish index of multiple deprivations<br>Maternal smoking status<br>Gestational age<br>Mode of delivery<br>Number of births<br>Congenital diseases<br>Length of hospital stay for the index admission<br>Birth weight<br>Apgar score at 5 minutes<br>Admission to neonatal unit after birth<br>Number of previous pregnancies<br>Breastfeeding status at the mother's discharge | Adjusted hazard ratio of first asthma and wheeze admission in 4 to <6 year olds (15,669 exposed and 7,696 controls)<br>• Age ≤ 5 months vs. no RSV hospitalization: 1.0 (95% CI: 0.6–1.8)<br>• Age 6–23 months vs. no RSV hospitalization: 1.9 (95% CI: 1.2–2.9)<br>Adjusted hazard ratio of first asthma and wheeze admission in 6 years or more (15,669 exposed and 7,696 controls)<br>• Age ≤ 5 months vs. no RSV hospitalization: 1.0 (95% CI: 0.6–1.8)<br>• Age 6–23 months vs. no RSV hospitalization: 1.5 (95% CI: 1.0–2.5) | First asthma and wheeze admission rate per 1000 child-years in 4 to <6 years (15,669 exposed and 7,696 controls)<br>• Age ≤ 5 months: 6.5 (95% CI: 5.1–8.1)<br>• Age 6–23 months: 7.2 (95% CI: 5.4–9.4)<br>First asthma and wheeze admission rate per 1000 child-years in 6 years or more (15,669 exposed and 7,696 controls)<br>• Age ≤ 5 months: 2.8 (95% CI: 2.1–3.6)<br>• Age 6–23 months: 3.5 (95% CI: 2.6–4.6) |

Abbreviation: *:RSV = respiratory syncytial virus, †:CI = confidence interval

defined RSV hospitalization before 2 years of age and asthma outcomes beginning after 2 years of age with an unclear interval between exposure and outcome. The study by Zhou et al. assumed a linear age-effect using continuous age from 0 to 24 months, however, it is possible that the effect of severe RSV infection on asthma risk may be non-linear as there may be some protection from maternal antibodies in the first 3 months of life and there may be different capacity for both the developing immune system and airway to respond to and to recover from RSV infection [18]. In addition, as age was categorized in four of the studies, whether there is variable directionality within these age ranges is not known. Ideally knowing whether the trends within the age categories were the same would be informative. This is particularly important, as most studies of severe RSV infection identify infant age between approximately 3–6 months as the peak risk for hospitalization, with younger and older ages on either side of this range generally associated with lower rates of RSV hospitalization [19]. Finally, since risk of RSV hospitalization decreases with advancing child age on a population level, the finding of an increased risk of asthma among children with advancing age of RSV hospitalization during the first 2 years of life could be identifying those children genetically predisposed to developing asthma, a demonstration of the problem of shared heredity in studies of the association of severe RSV infection and risk of asthma [20]. Lastly, as with all observational studies there could be unmeasured confounding, and based on the QUIPS tool these studies all had a high risk of bias.

### Informing future studies

This systematic review elucidated some limitations specific to study designs of assessing the association between age of RSV infection and asthma risk that may be overcome in designing future studies. First, although retrospective cohort studies, especially those using administrative databases have the benefit of including large samples, measurement errors of RSV infection and childhood asthma still need to be overcome. In daily practice, testing for RSV for mild upper respiratory tract infections is not routinely done as it does not generally change clinical management, highlighting the challenges of using existing healthcare data to define

RSV infection. Further, the diagnostic criteria of childhood asthma varied substantially across the studies included in this systematic review. As asthma hospitalization is an infrequent outcome among children with asthma, studies with asthma hospitalization as the outcome of interest likely underestimate asthma incidence and prevalence. This problem could be mitigated by the use of validated definitions for asthma, although we recognize that it was not the primary aim of each of these studies to determine the relationship between age of infection and asthma risk [21]. Second, as there is a shared genetic predisposition between severe RSV infection and childhood asthma inception, there may be differential effects of age of infection on asthma risk among those with and without a family history of asthma, as well as on the asthma phenotype that results [4, 20]. Thus, severe RSV infection or RSV hospitalization may be an early marker of asthma risk, and assessing age of severe events may not inform defining susceptible infant age of infection with a causal association with asthma. As we only measure proxies of genetic predisposition such as family members' asthma and/or allergic diseases, this can only partially correct for genetic confounding. A solution to genetic confounding is to use RSV infection rather than severe RSV infection requiring hospitalization, as there is unlikely genetic confounding between RSV infection and asthma [22]. Third, in the matched-pair cohort design, ideally the comparator group should be selected so that exposure status cannot change in the same subject. Wang et al. used children born during the same period and below 2 years of age with hospitalization due to unintentional accidents as the comparator. However, this group includes those infected and uninfected across this 2-year age period. Prospective studies could overcome many of these challenges; however, conducting RSV surveillance of a large population requires significant financial, time and human resources, and the sample size to assess the association of age/timing of RSV infection with sufficient granularity and asthma risk would likely need to be very large. An alternative could be to utilize a routinely available variable such as date of birth as an instrumental variable for RSV infection.

## Conclusions

This systematic review of the association between age of RSV infection and asthma risk could only identify studies of RSV-hospitalization and could not provide summary statistics nor valid conclusions. This systematic review highlights gaps in our knowledge of whether there is a susceptibility age range during which RSV infection confers the highest risk of later development of asthma and the challenges of designing studies to address this question. To determine the association between age of RSV infection and risk of asthma, a large-scale prospective cohort study will be an ideal study design to overcome measurement bias and confounding. Another option is an instrumental variable approach taking advantage of the well observed association between birth month and asthma [23]. It is unlikely a person's date of birth causes his/her risk for asthma, a formal instrumental analysis demonstrating that the date of birth and asthma association is through and only through increasing the risk of having RSV infection will provide strong evidence of the causal relationship. We are planning to conduct further studies based on these two approaches. This systematic review will hopefully renew interest and efforts to understand the age-dependent effect of early life exposures such as RSV infection on subsequent asthma risk, as this has important implications for considering and designing asthma prevention trials.

## Supporting information

**S1 Checklist. PRISMA 2020 checklist.**
(DOCX)

**S1 Table. Search terms in each database.** Abbreviations: *: RSV = respiratory syncytial virus.
(DOCX)

**S2 Table. Signaling questions for quality assessment.**
(DOCX)

**S3 Table. Patient characteristics.** *: The number of participants included in the analysis evaluating the association between age at first RSV infection and child asthma inception.
(DOCX)

**S4 Table. QUIPS tool quality assessment of included studies.**
(DOCX)

## Acknowledgments

We thank Rachel Lane Walden, MLIS at Eskind Biomedical Library at Vanderbilt University for assisting with the literature search.

## Author Contributions

**Conceptualization:** Akihiro Shiroshita, Tebeb Gebretsadik, Pingsheng Wu, Nejla Zeynep Kubilay, Tina V. Hartert.

**Data curation:** Akihiro Shiroshita, Tebeb Gebretsadik, Nejla Zeynep Kubilay, Tina V. Hartert.

**Formal analysis:** Akihiro Shiroshita, Tina V. Hartert.

**Investigation:** Akihiro Shiroshita, Tebeb Gebretsadik, Nejla Zeynep Kubilay, Tina V. Hartert.

**Methodology:** Akihiro Shiroshita, Tebeb Gebretsadik, Pingsheng Wu, Nejla Zeynep Kubilay, Tina V. Hartert.

**Project administration:** Akihiro Shiroshita, Tebeb Gebretsadik, Nejla Zeynep Kubilay, Tina V. Hartert.

**Resources:** Akihiro Shiroshita, Tebeb Gebretsadik, Tina V. Hartert.

**Software:** Akihiro Shiroshita, Tebeb Gebretsadik, Tina V. Hartert.

**Supervision:** Tebeb Gebretsadik, Pingsheng Wu, Nejla Zeynep Kubilay, Tina V. Hartert.

**Validation:** Tebeb Gebretsadik, Pingsheng Wu, Nejla Zeynep Kubilay, Tina V. Hartert.

**Visualization:** Akihiro Shiroshita, Pingsheng Wu, Nejla Zeynep Kubilay, Tina V. Hartert.

**Writing – original draft:** Akihiro Shiroshita, Tebeb Gebretsadik, Pingsheng Wu, Nejla Zeynep Kubilay, Tina V. Hartert.

**Writing – review & editing:** Akihiro Shiroshita, Tebeb Gebretsadik, Pingsheng Wu, Nejla Zeynep Kubilay, Tina V. Hartert.

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
