## [Decision Letter · Decision Letter 0]

28 Nov 2023

PONE-D-23-34011Association between age of respiratory syncytial virus infection and childhood asthma: A systematic reviewPLOS ONE

Dear Dr. Shiroshita,

Thank you for submitting your manuscript to PLOS ONE. After careful consideration, we feel that it has merit but does not fully meet PLOS ONE’s publication criteria as it currently stands. Therefore, we invite you to submit a revised version of the manuscript that addresses the points raised during the review process.

We look forward to receiving your revised manuscript.

Kind regards,

Dong Keon Yon, MD, FACAAI, FAAAAI

Academic Editor

PLOS ONE

Journal Requirements:

"AS received financial support for his doctoral study from Vanderbilt University Medical Center, Center for Asthma Research and the Fulbright Association, TG is supported by grants from the NIH, TH is supported by grants from the NIH and serves on DSMBs for Pfizer, and as an external scientific consultant for Sanofi."

Reviewers' comments:

Reviewer's Responses to Questions

**Comments to the Author**

1. Is the manuscript technically sound, and do the data support the conclusions?

Reviewer #1: Yes

Reviewer #2: Yes

Reviewer #3: Yes

2. Has the statistical analysis been performed appropriately and rigorously? 

Reviewer #1: Yes

Reviewer #2: Yes

Reviewer #3: Yes

3. Have the authors made all data underlying the findings in their manuscript fully available?

Reviewer #1: Yes

Reviewer #2: Yes

Reviewer #3: Yes

4. Is the manuscript presented in an intelligible fashion and written in standard English?

Reviewer #1: Yes

Reviewer #2: Yes

Reviewer #3: Yes

5. Review Comments to the Author

Reviewer #1: This is a good systemic review which highlights about the relationship between age of RSV infection and asthma risk. Due to the heterogeneous epidemiological designs of the enrolled studies. Although this study could not provide summary statistics nor valid conclusions, it still worth to know by journal readers. Furthermore, a large-scale prospective cohort study or an instrumental variable approach with a valid variable for RSV infection are required to overcome measurement bias and confounding.

Reviewer #2: It is an interesting study, since it investigates whether there is clear information about the age at which RSV occurs and not hospitalization. I liked the emphasis given to it, it opens up fields of research that have always been suggested, but there are no quality studies.

Reviewer #3: The current study highlights gaps in our knowledge of whether there is a susceptibility age range during which RSV infection confers the highest risk of later development of asthma and the challenges of designing studies to address this question. The authors assume a large-scale prospective cohort study to address these gaps. However, in my personal opinion, a large-scale prospective cohort as you designed is difficult to implement. Can you design a simple, highly feasible observational study with a relatively simple test method and without high risk of bias to determine the association between RSV infection and asthema?

Minor aspects: 1. page 17, lines 236, 237, a repeated sentence “Compared to those without RSV hospitalization”.

2.Figure 1, in my opinion, do not draw the flow diagram by yourself, just use the original PRISMA flow diagram by filling your contents.

6. PLOS authors have the option to publish the peer review history of their article (what does this mean?). If published, this will include your full peer review and any attached files.

Reviewer #1: No

Reviewer #2: **Yes: **Lydiana Avila

Reviewer #3: **Yes: **Haoran Wang

---

## [Author Response · Author response to Decision Letter 0]

4 Dec 2023

Point-by-Point Responses to the Comments of the Reviewers

Academic Editor:

Response: We have revised the manuscript and confirmed it meets PLOS ONE’s style requirements.

"AS received financial support for his doctoral study from Vanderbilt University Medical Center, Center for Asthma Research and the Fulbright Association, TG is supported by grants from the NIH, TH is supported by grants from the NIH and serves on DSMBs for Pfizer, and as an external scientific consultant for Sanofi."

Response: We have added the statement “T In the Competing Interests section, this does not alter our adherence to PLOS ONE policy on sharing data and materials. In addition, we have updated the competinginterests statement in the cover letter.

3. Please include captions for your Supporting Information files at the end of your manuscript, and update any in-text citations to match accordingly. Please see our Supporting Information guidelines for more information: http://journals.plos.org/plosone/s/supporting-information..

Response: We have revised the manuscript accordingly and confirmed that it adheres to the Supporting Information guidelines.

Reviewer #1: This is a good systemic review which highlights about the relationship between age of RSV infection and asthma risk. Due to the heterogeneous epidemiological designs of the enrolled studies. Although this study could not provide summary statistics nor valid conclusions, it still worth to know by journal readers. Furthermore, a large-scale prospective cohort study or an instrumental variable approach with a valid variable for RSV infection are required to overcome measurement bias and confounding.

Response: We appreciate the reviewer’s recognition of the important question addressed and the gap that this manuscript identifies.

Reviewer #2: It is an interesting study, since it investigates whether there is clear information about the age at which RSV occurs and not hospitalization. I liked the emphasis given to it, it opens up fields of research that have always been suggested, but there are no quality studies.

Response: We are grateful for your thoughtful comments. 

Reviewer #3: The current study highlights gaps in our knowledge of whether there is a susceptibility age range during which RSV infection confers the highest risk of later development of asthma and the challenges of designing studies to address this question. The authors assume a large-scale prospective cohort study to address these gaps. However, in my personal opinion, a large-scale prospective cohort as you designed is difficult to implement. Can you design a simple, highly feasible observational study with a relatively simple test method and without high risk of bias to determine the association between RSV infection and asthema?

Response: Thank you for your comment. Although a large-scale prospective cohort study is an ideal study design, we agree with your opinion that it is challenging to implement. Children’s birthday/birth month has been consistently shown to be associated with an increased risk of RSV infection and childhood asthma. We thus propose an instrumental variable approach with date of birth as an instrumental variable to RSV infection. An association between date of birth and asthma/wheeze via and only via RSV infection will provide strong evidence of a causal relationship between RSV infection and asthma. We have added these future study plans in our conclusion section

"To determine the association between age of RSV infection and risk of asthma, a large-scale prospective cohort study will be an ideal study design to overcome measurement bias and confounding. Another option is an instrumental variable approach taking advantage of the well observed association between birth month and asthma. It is unlikely a person’s birthday causes his/her risk for asthma, a formal instrumental analysis demonstrating that the date of birth and asthma association is through and only through increasing the risk of having RSV infection will provide strong evidence of the causal relationship. We are planning to conduct further studies based on these two approaches.” (Page 29, Line 348–351)

Minor aspects: 

1. page 17, lines 236, 237, a repeated sentence “Compared to those without RSV hospitalization”.

Response: Thank you for pointing it out. We have deleted the repeated phrase.

“The study by Wang et al. estimated the hazard ratios of first asthma hospitalization among those with RSV hospitalizations at age 0–5 months and 6–23, compared to those without RSV hospitalization. Compared to those without RSV hospitalization, Cchildren with RSV hospitalization at age 6–23 months had a significantly higher hazard ratio of asthma hospitalization during age 4–6 years”

2. Figure 1, in my opinion, do not draw the flow diagram by yourself, just use the original PRISMA flow diagram by filling your contents.

Response: We have updated the figure using the original PRISMA flow diagram.

Figure 1.

---

## [Decision Letter · Decision Letter 1]

18 Dec 2023

Association between age of respiratory syncytial virus infection and childhood asthma: A systematic review

PONE-D-23-34011R1

Dear Dr. Shiroshita,

We’re pleased to inform you that your manuscript has been judged scientifically suitable for publication and will be formally accepted for publication once it meets all outstanding technical requirements.

Kind regards,

Dong Keon Yon, MD, FACAAI, FAAAAI

Academic Editor

PLOS ONE

Additional Editor Comments (optional):

This is an excellent paper.

Reviewers' comments:

Reviewer's Responses to Questions

**Comments to the Author**

1. If the authors have adequately addressed your comments raised in a previous round of review and you feel that this manuscript is now acceptable for publication, you may indicate that here to bypass the “Comments to the Author” section, enter your conflict of interest statement in the “Confidential to Editor” section, and submit your "Accept" recommendation.

Reviewer #3: All comments have been addressed

2. Is the manuscript technically sound, and do the data support the conclusions?

Reviewer #3: Yes

3. Has the statistical analysis been performed appropriately and rigorously? 

Reviewer #3: Yes

4. Have the authors made all data underlying the findings in their manuscript fully available?

Reviewer #3: Yes

5. Is the manuscript presented in an intelligible fashion and written in standard English?

Reviewer #3: Yes

6. Review Comments to the Author

Reviewer #3: (No Response)

7. PLOS authors have the option to publish the peer review history of their article (what does this mean?). If published, this will include your full peer review and any attached files.

Reviewer #3: **Yes: **Haoran Wang

---

## [Editor Report · Acceptance letter]

4 Feb 2024

PONE-D-23-34011R1 

PLOS ONE

Dear Dr. Shiroshita, 

I'm pleased to inform you that your manuscript has been deemed suitable for publication in PLOS ONE. Congratulations! Your manuscript is now being handed over to our production team.

Kind regards, 

on behalf of

Dr. Dong Keon Yon 

Academic Editor

PLOS ONE